# The Behavioral Intention to Use Virtual Reality in Schools: A Technology Acceptance Model

**DOI:** 10.3390/bs14070615

**Published:** 2024-07-19

**Authors:** Silvia Puiu, Mihaela Tinca Udriștioiu

**Affiliations:** 1Department of Management, Marketing and Business Administration, Faculty of Economics and Business Administration, University of Craiova, 200585 Craiova, Romania; 2Department of Physics, Faculty of Sciences, University of Craiova, 200585 Craiova, Romania

**Keywords:** behavioral intention, virtual reality, education, technology acceptance model, task–technology fit

## Abstract

This paper presents the impact of using virtual reality (VR) in education and focuses on the main factors that affect the behavioral intention of using this new technology for the benefit of both professors and students. As part of our research methodology, we conducted a survey based on the Technology Acceptance Model and used partial least squares structural equation modelling. The results show that the intention to use VR technology in education is influenced by both usage satisfaction and the task–technology fit. The system quality of the VR technology proved to be important in influencing both the task–technology fit and perceived usefulness. The findings are useful for professors considering the adoption of new technologies that might improve the efficiency of learning for their students. Additionally, managers of educational institutions can use these insights to direct investments towards this type of innovation, which appeals to newer generations.

## 1. Introduction

The present paper investigates the use of VR technology in the educational setting for both professors and students. With the rapid decline of the attention span among younger generations [1,2], schools and universities are trying to find new ways to engage in a more appealing way with the students and maintain their focus during lectures. Many studies show the VR’s role in increasing concentration [3,4]. Still, authors such as Graeske and Sjöberg [5] (p. 76) emphasize that “VR technology offers many opportunities, but cannot exist on its own”. VR is a “simulation system that can create and experience the virtual world” [6] (p. 335), which makes this technology useful for education too, bringing “a new color to the field of education” [7] (p. 14). 

As with any other technology, there may be problems regarding its acceptance and its successful and efficient implementation in a specific context, such as the educational one we address in this paper. Davis [8] defined the Technology Acceptance Model (TAM) and emphasized the factors involved in the decision to continue to use technology. These were related to both external factors, such as the design or quality features of the technology, and internal factors, related to the way the user perceives the technology in terms of ease of use and usefulness. 

There are many authors [9,10,11,12,13] interested in applying this well-known model to various fields, including education. The results are helpful for better understanding the way people interact with new technologies and for shaping strategies meant to efficiently incorporate the use of technology in schools. Chocarro et al. [9] applied the model to analyze the willingness to use chatbots (as part of artificial intelligence development) in the educational process. The results are useful for those creating technological products or services, helping to ensure they are more easily accepted and integrated by the users. 

The COVID-19 pandemic and the increased use of online education have sparked interest in more research, providing more opportunities to test the TAM [10,11]. In this context, the results help managers in schools and universities to adjust the use of technologies and offer training [11] or alternatives that fit the needs of both students and their professors. Fussell and Truong [12] (p. 249) state that VR “expands educational opportunities” and creates a “dynamic learning environment”.

## 2. Literature Review and Hypotheses Development

To apply the TAM, we focused on eight variables: quality factors related to the VR technology (information quality, system quality, and service quality), task–technology fit, perceived ease of use and perceived usefulness, usage satisfaction, and continuous usage intention. 

### 2.1. Quality Factors Related to the VR Technology (Information Quality, System Quality, and Service Quality)

These factors are external factors that can influence the decision to continue to use technology, and they are an important part of the TAM [8]. In a study on VR technology, Du et al. [14] analyzed the information quality, the system quality, and the service quality. These three variables are widely used in other research papers [15,16,17,18,19,20]. Martono et al. [15] studied the influence of these factors on the employees’ perceived ease of use and usefulness of the information system and found a direct correlation. Tao [17] introduced information quality and system quality to describe the electronic resources used by students and analyzed their impact on the decision to use these types of resources. Information quality was introduced by Colvin and Goh [18] in their TAM to describe the use of computers by the police. Lin [19] studied these three factors to analyze their impact on the use of information systems by nurses in Taiwan. The information quality of the World Wide Web was used by Lederer et al. [20] (p. 269) to better understand the “decisions to revisit sites” that are “relevant” for the user. 

### 2.2. Task–Technology Fit

Goodhue and Thompson [21] proposed the model of task–technology fit (TTF) as a way to see the users’ performance through the lens of both the technology and the tasks that need to be accomplished by using that technology. A good fit between technology and tasks will lead to higher performance, acceptance, and continuous usage of that technology. TTF has been researched by many authors across different domains: banking [22,23], retail [24], sports [25], human resources [26], and education [27,28,29,30,31,32]. 

Masrom et al. [27] focused on the impact of task–technology fit on students using video-based learning. Albeedan et al. [28] used both TTF and TAM to understand the impact of mixed reality for students in police academies. Their results revealed a positive correlation between TTF and the usefulness perceived by the students. Jardina et al. [29] studied the correlation between TTF and the performance of both students and teachers in using electronic textbooks. Al-Emran [30] analyzed the impact of TTF on the usefulness of smartwatches in the educational process and found a positive correlation with student performance. Alyoussef [31] found a direct relationship between TTF and students’ performance and satisfaction when using e-learning. Howard et al. [32] noted that TTF “was a very strong moderator” for the use of VR in training. 

### 2.3. Perceived Ease of Use

In the TAM, perceived ease of use (PEU) is the subjective perspective of the users when interacting with a new technology. Many authors studied this factor to understand how the intention to use technology is shaped by it and also what other factors might influence the users’ perception. Fagan et al. [33] found a positive correlation between PEU and the intention of students in the USA to continue using VR. VR use and its relationship with PEU are studied in many areas beyond education, such as culture [34], consumer behavior [35], or tourism [36]. 

The interest in studying the benefits of using VR in education and training has increased in recent years, especially after the start of the COVID-19 pandemic and its restrictions. which shifted traditional education to e-learning and led to the incorporation of new, dynamic, and appealing technologies for younger generations [37,38,39]. Alqahtani et al. [38] found that the intention to use VR and e-learning positively influenced the students’ perception of the easiness of incorporating the technology. In another study, Sagnier et al. [40] could not validate the impact of PEU on the intention to use VR because the correlation, similar to the quality factors, was not significant. 

Raja and Lakshmi Priya [41] revealed that PEU strongly influences the teachers’ decision to use VR in class because not all of them are familiar with the technology and not all of them learn in the same way. Therefore, if a technology seems easy to use, it will determine, in the end, whether that technology will be integrated or not into the educational setting. The authors also noted that VR and other new technologies are not so common in countries that are less economically developed. Huang and Liaw [42] identified that PEU is influenced by the users’ perceived self-efficacy. 

### 2.4. Perceived Usefulness

Ferdinand et al. [43] studied the impact of perceived usefulness (PUF) on the effectiveness of using VR technology to teach students in science and found a positive influence on their achievements. Wong et al. [44] focused on the relationship between PUF, VR effectiveness, and personal characteristics to understand the learning process for employees in corporations. The results revealed that PUF is dependent on the open-mindedness of the employees, suggesting that those who are open to trying new technology and learning new procedures at work using VR also perceived it as more useful and, in the end, more efficient because they were also “more engaged” [44] (p. 2149). Jo and Park [45] researched the factors influencing PUF, and “affordance” was the one that had a direct impact on both PUF and “enjoyment” of using the technology. Du et al. [14] noticed that PUF is mainly influenced by quality factors, but the authors point to the fact that usually, professors are also more open to these new technologies. Sultan et al. [46] found that most medical students in their study appreciated the usefulness of VR technology. 

### 2.5. Usage Satisfaction

Osmani [47] researched medical students’ satisfaction after using VR during the COVID-19 pandemic, and the results revealed an average level, which was also influenced by their computer skills. A study conducted on Indian students [48] showed satisfaction in using VR for learning purposes, influencing its continuous usage. Gim et al. [49] (p. 279) highlight that usage satisfaction is influenced by both “self-determination” (a factor related to individual characteristics) and quality factors related to VR technology. The usage satisfaction factor was above average in a study conducted on medical students using VR [46]. Fink et al. [50] studied the use of “VR field trips” and found a high level of satisfaction among students, which was influenced by the perceived usefulness of the technology. The students’ competencies influence their satisfaction with using VR in education [51]. 

### 2.6. Continuous Usage Intention

Du and Liang [52] studied the use of VR by professors in China. They identified the factors with a great impact on the continuous usage intention (CUI): “performance expectancy, effort expectancy, social influence, facilitating conditions, and hedonic motivation”. Salloum et al. [53] highlight the role of “perceived complexity and perceived enjoyment” in continuing to use the metaverse in education. Zhang et al. [54] identified that perceived usefulness and usage satisfaction are the factors mostly influencing CUI. In another study on Taiwanese students, a direct correlation was noticed between “performance expectancy”, “effort expectancy”, “social influence” and “facilitating conditions” and CUI [55]. Personal characteristics, such as the professors’ openness and their digital skills were also found to impact CUI [56]. Chen et al. [57] state that the intention to use VR was mostly influenced by perceived usefulness, usage satisfaction, and students’ self-efficacy.

Following the literature review, we established the following hypotheses, which address the main questions of our research, mainly to identify the factors that influence the decision to use VR technology for teaching and learning. The factors are related to the technology (the quality factors and the task–technology fit), but also to the individual who uses the technology (their perceived ease of use and usefulness, and their satisfaction in using the VR technology). 

**Hypothesis 1 (H1).** 
*The quality factors (information, service, and system) of VR technology directly and positively influence the users’ perceived ease of use.*


**Hypothesis 2 (H2).** 
*The quality factors (information, service, and system) of VR technology directly and positively influence the users’ perceived usefulness.*


**Hypothesis 3 (H3).** 
*The quality factors (information, service, and system) of VR technology directly and positively influence the task–technology fit.*


**Hypothesis 4 (H4).** 
*The users’ perceived ease of use directly and positively influences their usage satisfaction.*


**Hypothesis 5 (H5).** 
*The users’ perceived usefulness directly and positively influences their usage satisfaction.*


**Hypothesis 6 (H6).** 
*The task–technology fit directly and positively influences the continuous usage intention.*


**Hypothesis 7 (H7).** 
*Usage satisfaction directly and positively influences the continuous usage intention.*


## 3. Research Methodology

We applied the model by using a survey sent to students and professors in Romania who had previously tested VR technology in an educational setting. The respondents previously tested VR headsets and several applications (space simulators, anatomy apps, science apps, and virtual tours in factories, hospitals, universities, laboratories, and campuses).

The survey included statements on a Likert scale from 1 (total disagreement) to 5 (total agreement), replicating the study conducted by Du et al. [14]. We extended their study to include both students and professors. To analyze the results, we used partial least squares structural equation modelling (PLS-SEM) and SmartPLS version 4 [58]. The main advantage of PLS-SEM is its capability to be used with smaller samples, which is especially important when studying a technology that is not yet extensively used in education. The limitations can be related to the sample size and the input of data regarding the respondents’ perceptions of the use of VR technology. 

We collected 156 answers between November 2023 and February 2024: 60% were students and 40% professors. The encountered difficulties were related to the fact that VR technology is quite new in Romania, particularly in educational contexts, and the survey could be filled out only after previously using the technology, thus requiring being able to understand its benefits and challenges. 

Figure 1 depicts the Technology Acceptance Model used to test the following variables: information quality—INFQ (with three items), system quality—SYSQ (with three items), service quality—SRVQ (with two items), perceived ease of use—PEU (with three items), perceived usefulness—PUF (with three items), task–technology fit—TTF (with three items), usage satisfaction—USTF (with two items), and continuous usage intention—CUI (with three items). All the constructs and items of the research model are detailed in Table 1. 

## 4. Results

We applied the PLS-SEM algorithm to determine the convergent validity of the Technology Acceptance Model for the use of VR in education. Thus, we determined the Variance Inflation Factor (VIF) and the outer loadings for each item in the model, as can be seen in Table 2. All outer loadings are higher than 0.8, and all VIF values are below 5, which ensures the high convergent validity of the items in the model and the desired low collinearity [59]. 

As shown in Figure 2, the strongest impact between the constructs in the model was from SYSQ to TTF (0.670), from USTF to CUI (0.588), from SYSQ to PUF (0.559), from PEU to USTF (0.508), and from INFQ to PEU (0.476). The variance in CUI is explained in a proportion of 83.1% by the influence of USTF and TTF. The three quality variables (INFQ, SYSQ, and SRVQ) explain 74.8% of the variance in PUF, 71.7% of the variance in TTF, and 66.5% of the variance in PEU.

In Table 3, we present the values for Cronbach’s alpha, composite reliability, and average variance extracted (AVE) for the model’s constructs. All Cronbach’s alpha, composite reliability, and AVE values are above 0.8, which ensures a high level of reliability, validity, and consistency of the model. The model fit is also indicated by the value of SRMR, which is 0.041, lower than the recommended threshold of 0.08 [58]. 

In Table 4, we present the Fornell–Larcker criterion, which shows the discriminant validity of the model. The values in the main diagonal (the square roots of the average variance extracted) are higher than the values in the same column, which measure the relation with the other constructs in the model. 

In Table 5, we present the results after applying the bootstrap test for a 5% significance level. The t-values below 1.96 and p values higher than 0.05 indicate that the hypothesis was not validated, in which case the confidence interval includes the zero value. 

## 5. Discussion

Perceived ease of use was influenced directly and positively by the information quality of VR technology but not by the service quality and system quality. Meanwhile, perceived usefulness was influenced by both information quality and system quality but not by service quality. H1 was partially validated (only the information quality has a direct and positive influence on perceived ease of use), which is in accordance with other studies [15,16]. In contrast, Du et al. [14] did not find a significant correlation between information quality and perceived ease of use, but they found a correlation between both system quality and service quality and perceived ease of use. H2 was partially validated (only information quality and system quality influenced perceived usefulness), and the finding is similar to those reached by other authors [14,15,16,17,20]. 

Our results did not show a significant impact of information quality and service quality on task–technology fit. Instead, system quality was proven to generate a direct and positive correlation with task–technology fit. Thus, H3 was partially validated. Du et al. [14] researched and validated the impact of all three quality factors on task–technology fit. In our study, H6 was validated, indicating the significant influence of task–technology fit on continuous usage intention. These findings were also validated by other researchers [14,22,60]. This correlation emphasizes the importance of training professors to use new technologies (such as VR) in education in a way that helps students accomplish the tasks they are given. Only in this way, they will be motivated to keep using the technology for learning. 

H4 was validated, demonstrating a direct and positive correlation between perceived ease of use and usage satisfaction. This result is in line with other researchers’ findings [14,61]. Students and professors using VR in an educational setting are more satisfied if they perceive the technology to be easy to use. This finding is helpful for managers in schools and universities to provide proper training and facilitate the integration of new and modern technologies in the classroom. H5 was validated, indicating a direct and positive correlation between perceived usefulness and usage satisfaction, as revealed by other authors discussed in the literature review [14,48,61]. This highlights that not only the easiness of use but also the usefulness of VR technology determine the level of satisfaction among students and professors. 

In our study, H7 was validated, showing a direct and positive correlation between usage satisfaction and continuous usage intention, in accordance with findings in other papers [48,54,57]. Still, this hypothesis was not validated by Du et al. [14]. Our results point to the need to ensure that the users are satisfied (by making the technology easy to use and highlighting its usefulness); otherwise, they will not continue using VR technology for teaching and learning.

## 6. Conclusions

The research focused on the use of VR in an educational context and found that quality factors related to the technology (and especially those related to information) influence perceived ease of use and usefulness. Perceived ease of use and perceived usefulness influence users’ satisfaction with VR, whereas usage satisfaction and task–technology fit directly impact the intention to further use the technology for educational purposes. 

The theoretical and practical implications are that these findings are useful for managers, professors, and other decision-makers in educational institutions to develop better strategies for education, integrating successfully newer technologies that can increase students’ academic performance. The advancement of technology is a reality that should be embraced by professors. They can use new technologies, such as virtual reality, to help their students understand abstract concepts (it can be successfully used by medicine students or science students), allowing them to practice specific behaviors in simulation applications designed for VR headsets (space simulations, health-related simulations, etc.). Applying these technologies in an educational setting requires training for professors to manage and integrate them into the lectures, seminars, and laboratories as additional tools for traditional learning. This translates into higher costs for educational institutions, and managers should consider the long-term potential of these investments to increase the academic performance of their students and to make teaching and learning more efficient. 

Limitations of this study are related to the sample size. The respondents tested the VR for a limited time, the technology not being used on a larger scale in Romania. After using it longer, there might be other factors that can be integrated into our future research directions, such as personal innovativeness [33,40] or cybersickness generated by prolonged use [40,62,63]. 

The results of our study and other papers in the literature review show the important potential that new technologies, such as VR, can offer for education. When implemented effectively, these technologies do not replace traditional education but complement it, enriching users’ experience, increasing satisfaction, and raising academic performance. 

## Figures and Tables

**Figure 1 behavsci-14-00615-f001:**
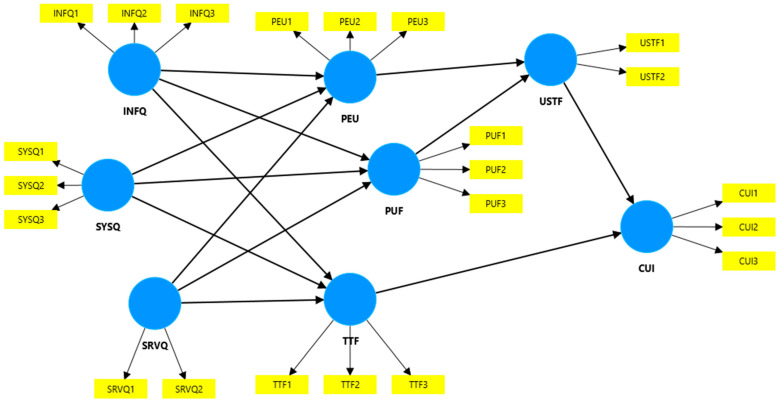
The research model—Technology Acceptance Model for the VR use in education.

**Figure 2 behavsci-14-00615-f002:**
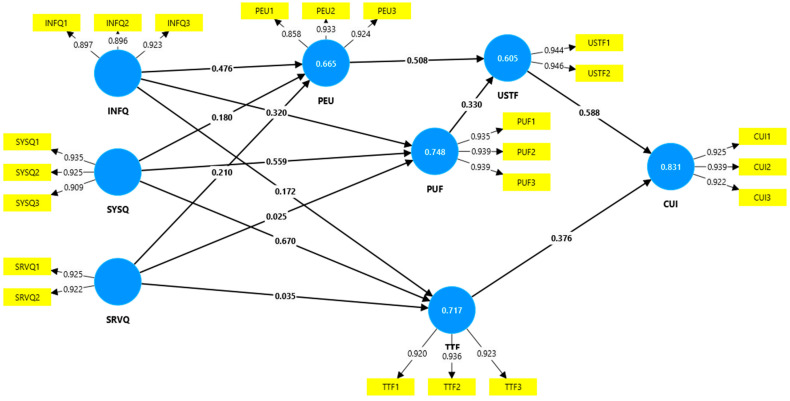
The path coefficients between the constructs in the model.

**Table 1 behavsci-14-00615-t001:** The constructs and items of the model regarding VR use in education.

Constructs	Items	Codes
Information quality of VR technology (INFQ)	The navigation through the menu of the VR headset was easy and smooth.	INFQ1
	The VR environment was very close to the reality.	INFQ2
	There were no crashes during the use of VR technology.	INFQ3
System quality of VR technology (SYSQ)	The VR system ensures a high level of interactivity in education.	SYSQ1
	The VR system optimizes the teaching and learning process.	SYSQ2
	The VR system offers an immersive teaching and learning experience.	SYSQ3
Service quality of VR technology (SRVQ)	There is access to knowledge related to VR use in education.	SRVQ1
	There is access to VR training for a better use of the system.	SRVQ2
Perceived ease of use for VR technology (PEU)	It was easy for me to use the VR technology.	PEU1
	Integrating VR technology into the educational process can be done easily.	PEU2
	Organizing an educational setting using VR technology is easy.	PEU3
Perceived usefulness of the VR technology in education (PUF)	VR technology stimulates the interest of students in learning.	PUF1
	VR technology is useful in understanding more abstract notions.	PUF2
	VR technology helps with the improvement of educational efficiency.	PUF3
Task–technology fit for VR use in education (TTF)	VR technology ensures a wide use of opportunities in education.	TTF1
	VR technology ensures the possibility of creating a teaching environment using simulation.	TTF2
	VR technology offers the possibility to better understand abstract concepts.	TTF3
Usage satisfaction with the VR technology (USTF)	I am satisfied with the overall experience of using VR technology.	USTF1
	I am satisfied with the efficiency of using VR for teaching and learning.	USTF2
Continuous usage intention regarding VR technology in education (CUI)	I am interested in continuing to use VR for educational purposes.	CUI1
	I intend to recommend VR technology to others.	CUI2
	I am interested in using VR technology as a standalone tool in education.	CUI3

**Table 2 behavsci-14-00615-t002:** The outer loadings and the collinearity of the model.

Items	Outer Loadings	VIF
CUI1	0.925	3.241
CUI2	0.939	3.773
CUI3	0.922	3.217
INFQ1	0.897	2.613
INFQ2	0.896	2.389
INFQ3	0.923	3.000
PEU1	0.858	1.990
PEU2	0.933	3.710
PEU3	0.924	3.489
PUF1	0.935	3.655
PUF2	0.939	4.006
PUF3	0.939	3.824
SRVQ1	0.925	1.994
SRVQ2	0.922	1.994
SYSQ1	0.935	3.601
SYSQ2	0.925	3.298
SYSQ3	0.909	2.762
TTF1	0.920	3.039
TTF2	0.936	3.672
TTF3	0.923	3.242
USTF1	0.944	2.612
USTF2	0.946	2.612

**Table 3 behavsci-14-00615-t003:** The constructs’ reliability and validity.

	Cronbach’s Alpha	Composite Reliability (rho_a)	Composite Reliability (rho_c)	AVE
CUI	0.920	0.921	0.950	0.863
INFQ	0.890	0.892	0.932	0.820
PEU	0.889	0.893	0.932	0.820
PUF	0.932	0.932	0.956	0.879
SRVQ	0.828	0.828	0.921	0.853
SYSQ	0.913	0.913	0.945	0.852
TTF	0.917	0.918	0.948	0.858
USTF	0.880	0.880	0.943	0.893

**Table 4 behavsci-14-00615-t004:** The discriminant validity—Fornell–Larcker criterion.

	CUI	INFQ	PEU	PUF	SRVQ	SYSQ	TTF	USTF
CUI	0.929							
INFQ	0.780	0.905						
PEU	0.776	0.795	0.905					
PUF	0.729	0.806	0.711	0.938				
SRVQ	0.698	0.804	0.732	0.716	0.924			
SYSQ	0.800	0.832	0.739	0.845	0.775	0.923		
TTF	0.833	0.757	0.713	0.829	0.692	0.840	0.926	
USTF	0.880	0.752	0.742	0.691	0.656	0.725	0.778	0.945

**Table 5 behavsci-14-00615-t005:** Bootstrapping test and hypotheses testing.

	T Statistics	*p* Values	Confidence Interval Bias-Corrected	Hypotheses Testing
INFQ -> PEU	3.828	0.000	(0.232, 0.722)	H1 partially validated (only INFQ influences PEU)
SRVQ -> PEU	1.809	0.070	(−0.010, 0.438)	
SYSQ -> PEU	1.401	0.161	(−0.098, 0.415)	
INFQ -> PUF	2.831	0.005	(0.115, 0.548)	H2 partially validated (only INFQ and SYSQ influence PUF)
SRVQ -> PUF	0.221	0.825	(−0.193, 0.247)	
SYSQ -> PUF	4.443	0.000	(0.302, 0.792)	
INFQ -> TTF	1.638	0.101	(−0.022, 0.392)	H3 partially validated (only SYSQ influences TTF)
SRVQ -> TTF	0.361	0.718	(−0.149, 0.230)	
SYSQ -> TTF	7.297	0.000	(0.481, 0.844)	
PEU -> USTF	5.974	0.000	(0.328, 0.658)	H4 validated
PUF -> USTF	3.510	0.000	(0.156, 0.522)	H5 validated
TTF -> CUI	4.065	0.000	(0.205, 0.571)	H6 validated
USTF -> CUI	6.132	0.000	(0.383, 0.764)	H7 validated

## Data Availability

Data will be made available upon written request.

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
