# Peer review of "The Behavioral Intention to Use Virtual Reality in Schools: A Technology Acceptance Model"

_behavsci, 2024, doi:10.3390/bs14070615_

Round 1

Reviewer 1 Report

Comments and Suggestions for Authors

You can find my comments in the file attached. Thank you very much for letting me revise this article.

Author Response

Dear Reviewer,

We are deeply grateful for the time and effort you have dedicated to analyzing our article. Your comments and recommendations have been invaluable, and we truly appreciate your astute observations that have helped us enhance the quality of our manuscript.

We sincerely appreciate all valuable observations and suggestions, which gave us a different perspective on our approach. In the following, we highlight your concerns and our efforts to address them. In response to your constructive criticisms, we have made revisions, which—we believe—improved the readability of our manuscript.

The revisions are highlighted in red in the manuscript:

 Response to Reviewer 1 Comments

  1. Although English is not my mother language, I think the paper is well written but, for me, it presents a handicap: the large number of acronyms. I highly recommend that the authors find a way to make the reading easier. For example, use the minimum acronyms in the Discussion and when they explain the Conclusions.

Reply: As you and the other reviewers suggested, we have rewritten the Discussion and Conclusions with the full names of the variables, avoiding the abbreviations for better clarity and readability of our manuscript.

  1. On another note, I think the authors in the conclusion could go deeper about the practical implications of using virtual reality in education and the elements that teachers and professors should consider in the design and implementation of learning situations using this kind of technology according to the results. Conclusions should generally provide more than a summary of the results and information about the limitations.

Reply: The Conclusion section was expanded to focus more on the practical implications. Thank you for your recommendation. The following paragraph was added: The advancement of technology is a reality that should be embraced by professors. They can use new technologies such as virtual reality to help their students understand abstract concepts (can be successfully used by medicine students or science students) and also by practicing a specific behavior in simulation applications designed for VR headsets (space simulations, health-related simulations, etc.). The applicability in an educational setting requires training for professors in managing these technologies and integrating them into the lectures, seminars, and laboratories as an additional tool for traditional learning. This translates into higher costs for educational institutions and manager should consider the long-term potential of these investments to increase the academic performance of their students and make teaching and learning more efficient.

  1. The authors mention in the Methodology they replicated the study conducted by Du et al. (2022) whose participants were students, but including both students and professors. It can be read in the Methodology section that the participants are students and professors (60% and 40%), more students than professors. I wondered if the authors analyzed the data considering the two types of participants (separately) and if they knew if the professors’ answers were similar to those of the students. That is to say, if they thought if the professors’ answers could be influenced the results. Did the authors consider this factor when making the discussion? Were the results of the other studies mentioned in the discussion (that use the same model (TAM)) also conducted with different types of participants in the same study? Or were all participants students?

Reply: There were no significant differences between students and professors. The survey was sent privately to students and professors and the professors did not interfere in the way students replied. Unfortunately, comparing students and professors and using PLS-SEM would require more than 150 students and 150 professors. As we explained in the manuscript, this was difficult to do because VR is not sufficiently used or tested in educational institutions in Romania. Both students and professors were enthusiastic about the use of VR but as we explain in the Conclusions, the respondents tested the VR for a limited time, the technology not being used on a larger scale in Romania. After using it longer, there might be other factors that can be integrated into our future research directions such as personal innovativeness [33,40] or the cybersickness generated by prolonged use [40,62,63].

We are hopeful that in the future, the technology will be used on a larger scale and a comparison between professors and students would be more statistically significant. In this research, both students and professors were users of VR technology and the technology was something new, something they tested but which is not regularly used in the teaching/learning process.

Jardina et al. [29] studied a similar model (the Task-technology fit Model) focusing on both students and professors using electronic textbooks. Several studies are focusing only on students or only on professors. In our study, students and professors are testers of a new technology such as VR. Even if there might be differences, in this case, for Romania, the technology is at the beginning, especially regarding education. We are aware that this might be a limitation of our study and we included it in the Conclusion section, under limitations.

  1. Finally, in the Methodology section, we only know about the VR experience that the participants “previously tested VR technology in an educational setting”. I think the authors could describe the details of the VR experience that the participants experimented with to better illustrate what kind of technology is VR, the main topic of the article.

Reply: We added a paragraph at the beginning of the Methodology to explain the concept of the VR experience better:  The respondents previously tested VR headsets and several applications (space simulators, anatomy apps, science apps, and virtual tours in factories, hospitals, universities, laboratories, and campuses).

Thank you again for your valuable suggestions and recommendations that helped us improve the quality of our manuscript.

Reviewer 2 Report

Comments and Suggestions for Authors

I honestly have a problem with 13 different hypotheses that are interrelated in a very difficult manner to be clear.

I strongly suggest decreasing the number of hypotheses and making a more straightforward proposal for the field. The topic is hot; the presentation is not correspondent.

Comments on the Quality of English Language

It needs to have a general checking by an English native speaker

Author Response

Dear Reviewer,

We are deeply grateful for the time and effort you have dedicated to analyzing our article. Your comments and recommendations have been invaluable, and we truly appreciate your astute observations that have helped us enhance the quality of our manuscript.

We sincerely appreciate all valuable observations and suggestions, which gave us a different perspective on our approach. In the following, we highlight your concerns and our efforts to address them. In response to your constructive criticisms, we have made significant revisions, which—we believe—improved the readability of our manuscript. The revisions are highlighted in red in the manuscript.

Reply to Reviewer 2

  1. I honestly have a problem with 13 different hypotheses that are interrelated in a very difficult manner to be clear. I strongly suggest decreasing the number of hypotheses and making a more straightforward proposal for the field. The topic is hot; the presentation is not correspondent.

Reply: We reduced the number of hypotheses as you and other reviewers suggested, comprising similar variables, but without affecting the Technology Acceptance Model we used.

We now have 7 hypotheses focusing on the factors that influence the decision to use VR technology for teaching and learning. The factors are related to the technology (the quality factors and the task-technology fit), but also to the individual who uses the technology (his/her perceived ease of use and usefulness, and his/her satisfaction in using the VR technology).

All changes are in red in the manuscript.

Thank you again for your help in improving the quality of our manuscript.

  1. It needs to have a general check by an English native speaker.

Reply: Thank you for your advice and your valuable feedback. We sent the manuscript for corrections to a native English speaker.

Reviewer 3 Report

Comments and Suggestions for Authors

The issue or study is topical and appealing for the international readership. I share authors' opinion that virtual reality can not replace traditional education but to complement it and enrich the users’  experience, increase satisfaction and raise academic performance. Data is well presented in tables, with a neat referencing; visualisation of data is well done.  This manuscript has a proper reference to literature. The references are relevant to the topic and cover recent developments.

The paper is overloaded with abbreviations and hypotheses. The authors could focus in few main hypotheses reflecting results in conclusions. This would be useful if the author identifies strengths and weakness of the method used in the manuscript. The arguments and discussions of findings are coherent, balanced and compelling, still could be more expanded.

I suggest considering this paper for publication after minor revision.

Author Response

Dear Reviewer,

We are deeply grateful for the time and effort you have dedicated to analyzing our article. Your comments and recommendations have been invaluable, and we truly appreciate your astute observations that have helped us enhance the quality of our manuscript.

We sincerely appreciate your valuable observations and suggestions. In the following, we highlight your concerns and our efforts to address them. In response to your constructive criticisms, we have made revisions, which—we believe—improved the readability of our manuscript.

The revisions are highlighted in red in the manuscript.

Reply to Reviewer 3

The issue of study is topical and appealing to the international readership. I share the authors' opinion that virtual reality cannot replace traditional education but complement it and enrich the users’ experience, increase satisfaction, and raise academic performance. Data is well presented in tables, with neat referencing; visualization of data is well done.  This manuscript has a proper reference to literature. The references are relevant to the topic and cover recent developments.

Reply: Thank you for your appreciation.

  1. The paper is overloaded with abbreviations and hypotheses. The authors could focus on a few main hypotheses reflecting results in conclusions.

Reply: We restructured the number of hypotheses and now we have 7. We kept the variables but grouped them to improve readability while also preserving the integrity of the Technology Acceptance Model we used. We also used the variables without abbreviations in Discussion and Conclusions to make the text easier to understand and clearer. Thank you for your help.

All changes are in red in the manuscript.

  1. This would be useful if the author identifies the strengths and weaknesses of the method used in the manuscript.

Reply: In the section Methodology, we included a paragraph to explain its main advantage, but also the limits of using PLS-SEM. The paragraph is the following: The main advantage of PLS-SEM is related to the fact that can be used with smaller samples and this is especially important in studying a technology that is not yet extensively used in education. The limits can be related to the sample, and the input of data referring to the respondents’ perceptions regarding the use of the VR technology.

  1. The arguments and discussions of findings are coherent, balanced, and compelling, still could be expanded.

Reply: The Discussion section was restructured to better present the new reduced hypotheses (7 from 13) and we also used the full name, not the abbreviations for the variables in the model to make it more readable. At your and the other reviewers' suggestions, we expanded the section Conclusions where we focus more on the practical implications of the findings presented in Discussion. The following paragraph was added: The advancement of technology is a reality that should be embraced by professors. They can use new technologies such as virtual reality to help their students understand abstract concepts (can be successfully used by medicine students or science students) and also by practicing a specific behavior in simulation applications designed for VR headsets (space simulations, health-related simulations, etc.). The applicability in an educational setting requires training for professors in managing these technologies and integrating them into the lectures, seminars, and laboratories as an additional tool for traditional learning. This translates into higher costs for educational institutions and manager should consider the long-term potential of these investments to increase the academic performance of their students and make teaching and learning more efficient.

I suggest considering this paper for publication after minor revision.

Reply: Thank you again for your valuable suggestions and your effort in helping us improve the quality of our manuscript.

Reviewer 4 Report

Comments and Suggestions for Authors

Line 55 the authors mentioned that they focused on 8 TAM variables. However, in their description they only focused on 6 i.e. section 2.1 to 2.6. Having 13 hypothesis makes the paper hard to follow. In addition, the hypotheses statements have "...directly and positively influence..." ; I am not sure if the analysis done in this manuscript provided evidence for both direct and positive influence. The analysis suggests evidence for positive influence.

The discussion is hard to follow because of overuse of abbreviations and forced the reader to go back and forth to double check the meaning. For example, understanding the following sentence would mean revisiting earlier sections of the manuscript. PEU was influenced directly and positively by INFQ (H1) but not by SRVQ (H6) and 241 SYSQ (H9), meanwhile, PUF was influenced by both INFQ (H2) and SYSQ (H10) but not 242 by SRVQ (H7). I suggest that for major claims, the authors should write complete sentences making the claims clear and easier to follow. 

Comments on the Quality of English Language

Good

Author Response

Dear Reviewer,

We are deeply grateful for the time and effort you have dedicated to analyzing our article. Your comments and recommendations have been invaluable, and we truly appreciate your astute observations that have helped us enhance the quality of our manuscript.

We sincerely appreciate your valuable observations and suggestions. In the following, we highlight your concerns and our efforts to address them. In response to your constructive criticisms, we have made revisions, which—we believe—improved the readability of our manuscript.

The revisions are highlighted in red in the manuscript.

Reply to Reviewer 4

  1. Line 55 the authors mentioned that they focused on 8 TAM variables. However, in their description, they only focused on 6 i.e. sections 2.1 to 2.6.

Reply: Thank you for your observation. It helped us clarify the text. We added in the title of section 2.1 the three quality factors we refer to: information quality, system quality, and service quality. They are presented in the literature review together being the quality factors inherent to a specific technology and they are always included in the Technology Acceptance Model.

  1. Having 13 hypotheses makes the paper hard to follow. In addition, the hypotheses statements have "...directly and positively influence..." ; I am not sure if the analysis done in this manuscript provided evidence for both direct and positive influence. The analysis suggests evidence for positive influence.

Reply: Thank you for your recommendation that helped us in making the manuscript more readable. The hypotheses were reduced to 7 and the sections were rewritten to match the changes.

Using the PLS-SEM algorithm, we notice from Figure 2 that the path coefficients between variables are positive (plus values) and the direct and positive influence was shown (or not) by applying the Bootstrapping test. This is why Hypotheses 1-3 were only partially validated because the direct and positive influence of some of the quality factors could not be validated (Table 7). Table 7 was also rewritten to match the changes to the hypotheses.

  1. The discussion is hard to follow because of overuse of abbreviations and forced the reader to go back and forth to double check the meaning. For example, understanding the following sentence would mean revisiting earlier sections of the manuscript. PEU was influenced directly and positively by INFQ (H1) but not by SRVQ (H6) and 241 SYSQ (H9), meanwhile, PUF was influenced by both INFQ (H2) and SYSQ (H10) but not 242 by SRVQ (H7). I suggest that for major claims, the authors should write complete sentences making the claims clear and easier to follow. 

Reply: Thank you again for helping us improve the quality of our manuscript. Your suggestion was very useful for us. We followed your recommendation and now, the Discussion and Conclusions are rewritten to include the full name and not the abbreviations of the variables in the model. Also, we changed the text accordingly to match the reduced number of hypotheses.

Round 2

Reviewer 2 Report

Comments and Suggestions for Authors

I think the authors have answered my concerns, just minor typos and grammar issues

Comments on the Quality of English Language

I think the authors have answered my concerns, just minor typos and grammar issues